# Prevalence and causes of blindness and vision impairment among people 50 years and older in Nepal: A national Rapid Assessment of Avoidable Blindness survey

Sailesh Kumar Mishra[1], Ranjan Shah[1], Parikshit Gogate[2,3,4]*, Yuddha Dhoj Sapkota[5], Reeta Gurung[6], Mohan Krishna Shrestha[6], Islay Mactaggart[7], Ian McCormick[7], Brish Bahadur Shahi[8], Rajiv Khandekar[9], Matthew Burton[7]

1 Nepal Netra Jyoti Sangh, Kathmandu, Nepal, 2 Community Eye Care Foundation, Dr. Gogate's Eye Clinic, Pune, India, 3 Department of Ophthalmology, D. Y. Patil Medical College, Pimpri, Pune, India, 4 School of Health Sciences, Queens University, Belfast, United Kingdom, 5 International Agency for Prevention of Blindness, South East Asia Office, Kathmandu, Nepal, 6 Tilganga Eye Institute, Kathmandu, Nepal, 7 International Centre for Eye Health, London School of Hygiene & Tropical Medicine, London, United Kingdom, 8 Ministry of Social Development, Birendranagar, Karnali Province, Nepal, 9 Department of Ophthalmology & Visual Sciences, Faculty of Medicine, The University of British Columbia, Vancouver, Canada

* parikshitgogate@hotmail.com

**Data Availability Statement:** The datasets are owned by Nepal Netra Jyoti Sangh and are available to download for registered users of the

## Abstract

### Purpose

To determine the prevalence and causes of blindness and vision impairment among people 50 years and older in Nepal.

### Methods

We conducted seven provincial-level Rapid Assessment of Avoidable Blindness (RAAB) cross-sectional, population-based surveys between 2018–2021. Provincial prevalence estimates were weighted to give nationally representative estimates. Sampling, enumeration, and examination of the population 50 years and older were done at the province level following standard RAAB protocol.

### Results

Across seven surveys, we enrolled 33,228 individuals, of whom 32,565 were examined (response rate 98%). Females (n = 17,935) made up 55% of the sample. The age-sex-province weighted national prevalence of blindness (better eye presenting visual acuity <3/60) was 1.1% (95% confidence interval [CI] 1.0–1.2%), and any vision impairment <6/12 was 20.7% (95% CI 19.9–21.5%). The prevalence of blindness was higher in women than men (1.3% [95% CI 1.1–1.5%] vs 0.9% [95% CI 0.7–1.0%]). Age-sex weighted blindness prevalence was highest in Lumbini Province (1.8% [95% CI 1.3–2.2%]) and lowest in Bagmati Province (0.7% [95% CI 0.4–0.9%]) and Sudurpaschim Province (0.7% [95% CI 0.4–0.9%]). Cataract (65.2%) was the leading cause of blindness in our sample, followed by

Rapid Assessment of Avoidable Blindness (RAAB) repository via https://www.raab.world/country-profiles/nepal (registration is free). Sudurpashchim Province https://www.doi.org/10.17037/RAABDATASET.00000385; Gandaki Province https://www.doi.org/10.17037/RAABDATASET.00000377; Madhesh Province https://www.doi.org/10.17037/RAABDATASET.00000379; Koshi Province https://www.doi.org/10.17037/RAABDATASET.00000378; Bagmati Province https://www.doi.org/10.17037/RAABDATASET.00000360; Karnali Province https://www.doi.org/10.17037/RAABDATASET.00000361; Lumbini Province https://www.doi.org/10.17037/RAABDATASET.00000349. The license restricts use of the data for commercial purposes. This restriction has been imposed in accordance with the research ethics approved institutionally by LSHTM and by partner institutions conducting RAABs.

**Funding:** The author(s) received no specific funding for this work.

**Competing interests:** No authors have competing interests.

corneal opacity (6.4%), glaucoma (5.8%) and age-related macular degeneration (5.3%). Other posterior segment diseases accounted for 8.4% of cases. Cataract was also the leading cause of severe vision impairment (83.9%) and moderate vision impairment (66.8%), while refractive error was the leading cause of mild vision impairment (66.5%).

## Conclusion

The prevalence of blindness was higher among women than men and varied by province. The Lumbini and Madesh Provinces in the Terai (plains) region had higher prevalence of blindness than elsewhere. Cataract was the leading cause of blindness, severe vision impairment and moderate vision impairment while refractive error was the leading cause of mild vision impairment.

## Introduction

Nepal has a long history of conducting population-based surveys of vision impairment and blindness. The first national survey in 1981 found a best-corrected blindness prevalence (visual acuity <3/60 in the better eye) of 0.84% and moderate to severe vison impairment prevalence (visual acuity <6/18 and ≥3/60) of 1.85% in the population aged 10 years and older [1]. Cataract was the leading cause of blindness, accounting for 80% of avoidable cases [1]. Shortly after this study, the Nepal Netra Jyoti Sangh (NNJS) was set up as an umbrella organisation of non-governmental organisations, under the aegis of the Government of Nepal, to reduce blindness and vision impairment [2]. A network of primary eye care centres, staffed by ophthalmic assistants, was established and linked to secondary and tertiary eye care facilities with help from international developmental agencies. Nepal endorsed the World Health Organization–International Agency for Prevention of Blindness initiative "VISION 2020 The Right to Sight" and, in 2001, developed its first national eye health plan [3].

In 1995, population-based surveys were done in two zones (the fourteen first-level subnational administrative units of Nepal until 2015), where little change in the prevalence of blindness since 1981 was found [4]. After that, a subset of districts in each of the three zones were surveyed using a custom survey design in 2002 (n = 1) [5] and 2006 (n = 2) [6]. In 2008–2010, a series of ten Rapid Assessment of Avoidable Blindness (RAAB) surveys was done across the remaining eleven zones. The prevalence of blindness and vision impairment in the all-age population was estimated to have decreased from 0.84% in 1981 to 0.35% in 2011 (2011 all-age estimate extrapolated from estimate among people aged 50 years and older) [7].

The last forty years have seen enormous multi-sectorial efforts to improve eye health in Nepal. The number of ophthalmologists has increased from seven in 1981 to 364 in 2019, while the number of optometrists has increased from zero to 857 and ophthalmic assistants increased from zero to 1,246 in the same period [8]. The eye hospitals set up with NNJS collectively performed almost 350,000 cataract surgeries in 2019 (rising from 11,002 in 1995).

There has not been a national prevalence of blindness survey since 1981, and no district or provincial study has been conducted since 2010. RAAB is a relatively accurate, fast, and cost-effective method to estimate the prevalence of blindness in the population 50 years and older. This study aimed to conduct a new nationally representative RAAB, including seven separate surveys of each of seven newly formed provinces in the country. Here, we present vision impairment and blindness estimates for the country and provinces.

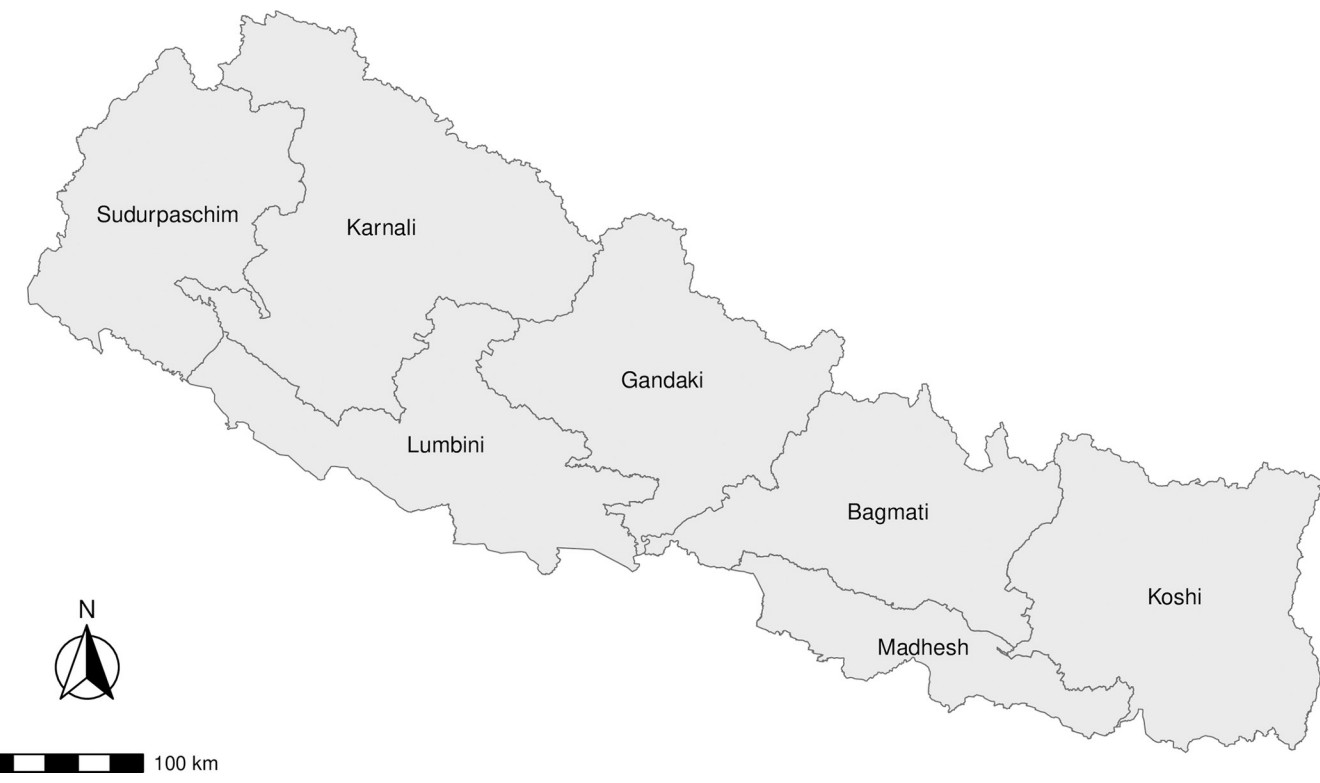

**Fig 1. Provincial-level map of Nepal.**

## Methods

Ethics approval was obtained from the Nepal Health Research Council under the Ministry of Health, Government of Nepal. The survey conformed to the tenets of the Declaration of Helsinki. Written informed consent was obtained from all participants before they enrolled in the study.

We conducted seven provincial-level cross-sectional, population-based surveys between 2018–2021: one Rapid Assessment of Avoidable Blindness survey per province (Fig 1). Together, the provincial-level sampling frames represented Nepal at the national level (Table 1).

### Sampling

Each provincial-level sample size calculation was based on an assumed prevalence of bilateral blindness in the population aged 50 years and older, using estimates from previous subnational RAAB surveys in Nepal done in 2008–2010, and the size of the population 50 years and older in that province according to the 2011 census (Table 1). Other parameters in the sample size calculations were consistent across the provinces: a relative precision of 20%, 95% confidence level, a design effect of 1.4 for a cluster size of 35 people and a 10% non-response rate. Sample size calculations were performed using the standard RAAB software.

RAAB uses a two-stage cluster random sampling methodology with same day enumeration and examination. We used wards (the lowest level population units defined in the 2011 census) as our primary sampling units. The total list of wards and their all-age population sizes (obtained from Central Bureau of Statistics, Ministry of Information and Technology) made

**Table 1. Sampling and data collection information for each provincial RAAB survey in Nepal 2018–2021.**

| Province | Sampling information | | | | Data collection | |
|---|---|---|---|---|---|---|
| | Estimated prevalence of blindness | Population 50 years and older (2011 census) | Sample size required | Number of clusters (size 35) | Data collection period | RAAB version used |
| Koshi | 3.3% | 743,329 | 4360 | 125 | Dec 2018—Mar 2020 | 6 |
| Madhesh | 3.5% | 760,911 | 4109 | 118 | Sept 2019—Feb 2020 | 7 |
| Bagmati | 2.5% | 831,208 | 5800 | 166 | June 2019—Dec 2019 | 7 |
| Gandaki | 3.0% | 399,059 | 4789 | 137 | Jan 2019—Feb 2020 | 6 |
| Lumbini | 3.0% | 655,927 | 4808 | 138 | July 2018—Dec 2018 | 6 |
| Karnali | 2.5% | 190,016 | 4067 | 117 | Jun 2019—Dec 2019 | 7 |
| Sudur Paschim | 2.9% | 250,982 | 4619 | 132 | Feb 2020—May 2021 | 7 |

up the sampling frame for each province. We selected the required number of clusters from wards with probability proportionate to their all-age population size. For the second stage sampling, we used cluster sketch-mapping and segmentation to enrol 35 participants per cluster. Based on the 2011 census, we estimated 15% of the all-age population was aged 50 years and older and that a population unit of approximately 235 people would contain 35 eligible people. If a selected population unit was smaller than 235 people, a second, adjacent ward in which to continue enrolment was pre-selected at random. If a selected population unit had a population greater than 470 it was divided into two or more segments and one segment was randomly selected by the ward chief or a local influencing leader. Once a segment boundary with approximately 35 eligible participants was defined, a starting corner was randomly chosen, and eligible people were enumerated by teams systematically moving household-to-household until 35 study participants were enrolled. If a selected segment did not have 35 eligible participants, enumeration continued in a pre-selected neighbouring population unit until 35 participants were enrolled. Data collection teams were accompanied by a local guide in every cluster.

## Eligibility criteria

All non-institutionalized people aged 50 years and older in Nepal were eligible for inclusion. To help verify age, the participant's birth year was compared to the death of King Mahendra and the start of the reign of King Birendra in 1972. Within a cluster, anyone resident for at least six months of the year was enrolled. Temporary visitors were excluded.

## Training

Before starting fieldwork, data collection teams attended a standardized training week run by a qualified RAAB trainer (YDS) in each province. Training included an interobserver variation exercise and a supervised pilot cluster. All teams achieved a Kappa score of 0.6 or greater for inter observer agreement on visual acuity, lens status, and cause of vision impairment.

## Survey time line

The ethics committees gave permission for a specific timeline for each province. The actual dates of the survey data collection are given in Table 1. The data collection was done for

Province 01 (Koshi) from 5December 2018 to March 2020; in Province 02 (Madhesh) from September 2019 to February 2020; Province 03 (Bagmati) from June 2019 to December 2019; in Province 04 (Gandaki) from June 2019 to February 2020; in Province 05 (Lumbini) from July 2018 to December 2018; in Province 06 (Karnali) from June 2019 to December 2019 and in Province 07 (Far western/ Sudur Paschim) from February 2020 to May 2021.

### Examination protocol

Examinations were conducted at participants' households on the same day as enrolment. We recorded the age and sex for all participants. We recorded distance and near spectacle ownership status and measured presenting distance visual acuity (VA) (i.e., with spectacles if available) in each eye. We measured pinhole VA for any eye where the presenting VA was worse than 6/12. VA was measured using tumbling E optotype cards (6/60, 6/18, and 6/12 sizes) outdoors at 6 meters, 3 meters, or 1 meter as required to test VA at 1/60, 3/60, 6/60, 6/18, and 6/12 thresholds. Four out of five correct responses were required to pass at each threshold. Perception of light was tested for any eye worse than 1/60. All eyes underwent a lens examination with a pen torch and distant direct ophthalmoscopy. Any eyes with presenting VA worse than 6/12 were assigned a cause of vision impairment from a standardised list and then a principal cause of vision impairment assigned per participant. The principal cause was determined as the cause more amenable to treatment or prevention. Eyes with presenting VA worse than 6/12 and no obvious anterior segment cause of vision impairment were dilated for fundus examination with direct ophthalmoscopy. Any participants with cataract and pinhole VA worse than 6/18 were asked about barriers to accessing treatment. Any participants who had received cataract surgery were examined further (type of surgery, reason for presenting VA worse than 6/12 if relevant) and asked about their surgical history (age at surgery, cost, place of surgery).

### Definitions

We defined vision impairment and blindness based on presenting visual acuity (i.e., with correction if available) in the better eye, according to the International Classification of Disease (ICD-11) thresholds [10]. Blindness was worse than 3/60, severe vision impairment (SVI) was worse than 6/60 but better than or equal to 3/60, moderate vision impairment (MVI) was worse than 6/18 but better than or equal to 6/60, and mild vision impairment was worse than 6/12 but better than or equal to 6/18.

### Data management

Data were collected on mobile Android devices. Three provinces used a mobile application developed for RAAB6 (mRAAB), and four used a beta version of the RAAB7 platform [9]. Data collected in mRAAB6 were converted to comma separated value (csv) files and emailed to survey coordinators upon completion of each cluster. Data were then uploaded to RAAB6 analysis software for processing (this process is now retired and no longer available). Data collected in the RAAB7 app were automatically synchronized to a secure Amazon Web Services cloud-based server when teams had internet connectivity [9].

### Analysis

A standardized epidemiological report was produced for each province using the RAAB7 automated analysis code available at https://github.com/raabteam/raab7-analysis. The analysis provided provincial age-sex weighted vision impairment prevalence estimates (post-stratified to the provincial 2021 Census population counts) and the principal causes of blindness and vision

impairment within the sample. To estimate a weighted national prevalence of vision impairment and blindness, each provincial dataset was exported from the RAAB software as a CSV file, appended in a single dataset, and imported into STATA 17. We used the svy command to account for the cluster survey design. We used unique cluster IDs to define the primary sampling unit (n = 952) and a provincial identifier to define strata (n = 7). To report age-sex-province weighted national estimates, we created a probability weight for fourteen 5-year age-sex categories (50–54, 55–59, 60–64, 65–69, 70–74, 75–79, 80+ for male and female). The probability weight was calculated by multiplying two weights: a provincial design weight (each of the seven provinces' proportion of the national population divided by 1/7, for male and female) multiplied by a sampling weight (the proportion of the sample in each 5-year age-sex group divided by the census proportion in the same age-sex group). The extrapolated magnitude of vision impairment was estimated by applying the age-sex-province weighted prevalence values reported in the paper to the 2021 census counts for female, male, and total populations separately; therefore, male and female estimates do not sum to total estimates.

This study is reported according to the relevant items in the STROBE checklist for cross-sectional studies [10].

## Results

We examined 32,565 participants of 33,228 enrolled (response rate 98%). Females made up 55% of the sample. The overall response rates in each province ranged from 95.4% to 99.5% (Table 2).

Nationally, the age-sex-province weighted prevalence of blindness among the population 50 years and older was 1.1% (95% CI 1.0–1.2%). We estimated that 60,138 people aged 50 years and older were blind in Nepal. The prevalence of blindness was significantly higher among females (1.3% [95% CI 1.1–1.5%]) than males (0.9% [95% CI 0.7–1.0%]); the difference in vision impairment prevalence by sex was not significantly higher at any other threshold of vision impairment (Table 3). The prevalence of any vision impairment <6/12 was 20.7% (95% CI 19.9–21.5%). Prevalence of any vision impairment <6/12 was higher among females (21.8% [95%CI 20.9–22.7%]) than males (19.5% [95% CI 18.6–20.5%]).

The provincial-level variation in blindness prevalence is shown in Fig 2A and Table 4. The overall prevalence of blindness was significantly higher in Lumbini Province (1.8% [95% CI 1.3–2.2%]) than in other provinces except Madhesh Province (1.4% [95% CI 1.0–1.8%]) and Karnali Province (1.0% [95% CI 0.6–1.3%]). More than half (52%) of the total extrapolated magnitude of blindness cases were in Madhesh and Lumbini Provinces (Table 4). The provincial level variation in the prevalence of any vision impairment <6/12 is shown in Fig 2B. Madesh Province (30.8% [95% CI 28.4–33.3%]) and Lumbini Province (27.3% [95% CI 25.0–

**Table 2. The response rate for each provincial RAAB and nationally.**

| Province | Female | | | Male | | | Total | | |
|---|---|---|---|---|---|---|---|---|---|
| | Enrolled n | Examined n | Examined % | Enrolled n | Examined n | Examined % | Enrolled n | Examined n | Examined % |
| Koshi | 2,266 | 2,221 | 98.0 | 1,965 | 1,900 | 96.7 | 4,231 | 4,121 | 97.4 |
| Madhesh | 2,168 | 2,158 | 99.5 | 1,907 | 1,897 | 99.5 | 4,075 | 4,055 | 99.5 |
| Bagmati | 3,331 | 3,204 | 96.2 | 2,479 | 2,338 | 94.3 | 5,810 | 5,542 | 95.4 |
| Gandaki | 2,618 | 2,583 | 98.7 | 2,176 | 2,134 | 98.1 | 4,794 | 4,717 | 98.4 |
| Lumbini | 3,104 | 3,082 | 99.3 | 2,538 | 2,500 | 98.5 | 5,642 | 5,582 | 98.9 |
| Karnali | 2,157 | 2,114 | 98.0 | 2,538 | 2,500 | 98.5 | 4,069 | 3,983 | 97.9 |
| Sudurpaschim | 2,603 | 2,573 | 98.9 | 2,004 | 1,992 | 99.4 | 4,607 | 4,565 | 99.1 |
| **National** | **18,247** | **17935** | **98.3** | **14,981** | **14630** | **97.7** | **33,228** | **32,565** | **98.0** |

**Table 3. Age-sex-province weighted estimates of blindness and vision impairment in Nepal.**

|  | Female | | | Male | | | Total | | |
|---|---|---|---|---|---|---|---|---|---|
|  | Weighted % | 95% CI | Extrapolated magnitude | Weighted % | 95% CI | Extrapolated magnitude | Weighted % | 95% CI | Extrapolated magnitude |
| **Blind** | 1.3 | 1.1–1.5 | 36,311 | 0.9 | 0.7–1.0 | 24,066 | 1.1 | 1.0–1.2 | 60,138 |
| **Severe VI** | 1.9 | 1.7–2.1 | 53,070 | 1.5 | 1.3–1.8 | 40,109 | 1.7 | 1.5–1.9 | 92,941 |
| **Moderate VI** | 7.9 | 7.4–8.5 | 220,659 | 7.2 | 6.7–7.8 | 192,525 | 7.6 | 7.2–8.0 | 415,501 |
| **Mild VI** | 10.7 | 10.1–11.3 | 298,868 | 10.0 | 9.4–10.6 | 267,396 | 10.3 | 9.9–10.8 | 563,113 |

### A. Age–sex weighted prevalence of blindness in Nepal by province (%)

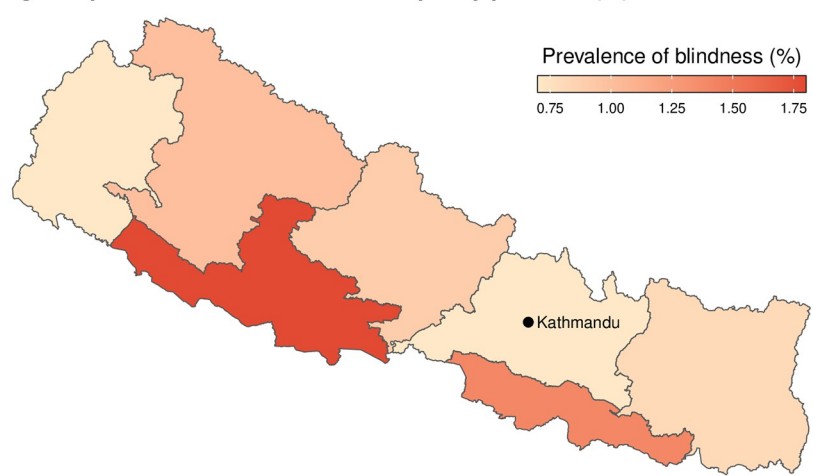

### B. Age–sex weighted prevalence of any vision impairment <6/12 by province (%)

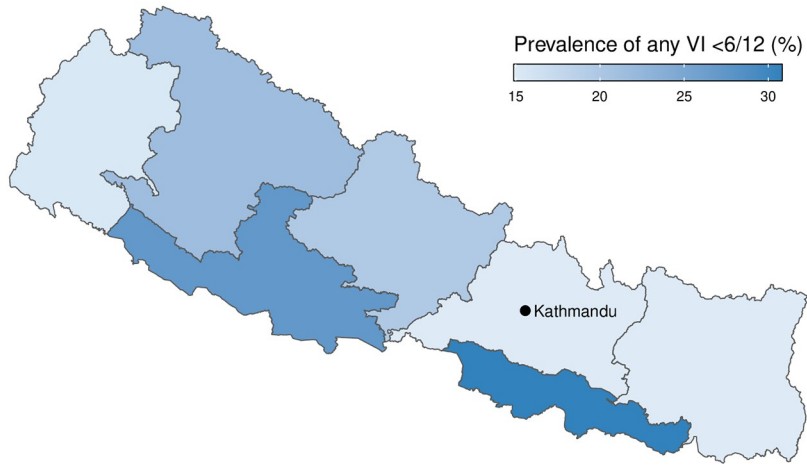

**Fig 2. Age-sex weighted prevalence of blindness and any vision impairment worse than 6/12 in Nepal by province.**

**Table 4. Age-sex weighted prevalence of bilateral blindness by province.**

| Province | Female | | | Male | | | Total | | |
|---|---|---|---|---|---|---|---|---|---|
| | Weighted % | 95% CI | Extrapolated magnitude | Weighted % | 95% CI | Extrapolated magnitude | Weighted % | 95% CI | Extrapolated magnitude |
| Koshi | 0.9 | 0.4–1.3 | 4,581 | 0.8 | 0.4–1.2 | 4,017 | 0.8 | 0.5–1.1 | 8,598 |
| Madhesh | 1.7 | 1.0–2.3 | 8,017 | 1.1 | 0.6–1.7 | 5,943 | 1.4 | 1.0–1.8 | 13,960 |
| Bagmati | 1.0 | 0.6–1.3 | 5,998 | 0.4 | 0.0–0.8 | 2,384 | 0.7 | 0.4–0.9 | 8,382 |
| Gandaki | 1.2 | 0.7–1.7 | 3,736 | 0.6 | 0.2–1.0 | 1,614 | 0.9 | 0.6–1.2 | 5,351 |
| Lumbini | 2.1 | 1.6–2.7 | 9,939 | 1.4 | 0.8–2.1 | 6,323 | 1.8 | 1.3–2.2 | 16,262 |
| Karnali | 0.9 | 0.4–1.3 | 1,144 | 1.1 | 0.6–1.6 | 1,372 | 1.0 | 0.6–1.3 | 2,517 |
| Sudur Paschim | 0.8 | 0.4–1.1 | 1,903 | 0.6 | 0.3–1.0 | 1,284 | 0.7 | 0.4–0.9 | 3,187 |

29.7%]) had significantly higher prevalence estimates than the other five provinces (S1 Appendix).

Within the nationwide sample, cataract was the leading cause of blindness (65.2%), followed by non-trachomatous corneal opacity (6.1%), glaucoma (5.8%), and age-related macular degeneration (5.3%). Cataract surgical complications caused 1.8% of blindness and 4.6% of moderate visual impairment. Diabetic retinopathy was 0.8% of blindness cases nationally, while 'other posterior segment conditions' contributed 8.4% of blindness cases (Table 5). Cataract was also the leading cause of severe vision impairment (83.9%) and moderate vision impairment (66.8%), while refractive error was the leading cause of mild vision impairment (66.5%) (Table 5).

## Discussion

Here we report the first nationally representative estimates of vision impairment and blindness, and their causes, in Nepal since 1981 [1]. This study was designed as a series of seven RAAB surveys representative of the population in each province, as well as the national population. The age-sex-province weighted national prevalence of blindness was 1.1% among the population 50 years and older.

**Table 5. The causes of blindness and vision impairment in the national sample.**

| | Blindness | | Severe VI | | Moderate VI | | Mild VI | |
|---|---|---|---|---|---|---|---|---|
| | N | % | n | % | n | % | n | % |
| **Refractive error** | 1 | 0.3 | 12 | 2.0 | 534 | 19.7 | 2,358 | 66.5 |
| **Aphakia** | 1 | 0.3 | 2 | 0.3 | 3 | 0.1 | 2 | 0.1 |
| **Cataract** | 257 | 65.2 | 501 | 83.9 | 1,810 | 66.8 | 953 | 26.9 |
| **Cataract Surgical Complications** | 7 | 1.8 | 11 | 1.8 | 126 | 4.6 | 79 | 2.2 |
| **Trachomatous Corneal Opacity** | 1 | 0.3 | 0 | 0.0 | 0 | 0.0 | 0 | 0.0 |
| **Other Corneal Opacity** | 24 | 6.1 | 7 | 1.2 | 35 | 1.3 | 31 | 0.9 |
| **Pterygium** | 0 | 0.0 | 0 | 0.0 | 9 | 0.3 | 8 | 0.2 |
| **Phthisis** | 7 | 1.8 | 1 | 0.2 | 4 | 0.1 | 5 | 0.1 |
| **Glaucoma** | 23 | 5.8 | 11 | 1.8 | 20 | 0.3 | 12 | 0.3 |
| **Diabetic Retinopathy** | 3 | 0.8 | 3 | 0.5 | 16 | 0.3 | 11 | 0.3 |
| **AMD** | 21 | 5.3 | 19 | 3.2 | 64 | 2.4 | 39 | 1.1 |
| **Myopic degeneration** | 3 | 0.8 | 4 | 0.7 | 11 | 0.2 | 6 | 0.2 |
| **Other posterior segment** | 33 | 8.4 | 22 | 3.7 | 70 | 1.0 | 36 | 1.0 |
| **All other globe/CNS** | 13 | 3.3 | 4 | 0.7 | 8 | 0.2 | 7 | 0.2 |
| **Total** | 394 | 100.0 | 597 | 100.0 | 2,711 | 100.0 | 3,547 | 100.0 |

**Table 6. Comparison of blindness prevalence in Nepal and other neighbouring countries.**

| Survey location | Year | Sample size | Age range | Blindness prevalence (definition) | Main cause | Per Capita | Author (reference) |
|---|---|---|---|---|---|---|---|
| Nepal | 1998 | 4,600 | 45 +yr | 5.3% (legal blind) | Cataract (78%) | | Pokharel [4] |
| Bangladesh | 2016 | 21,596 | | 2.2% (WHO) | Cataract (75.8%) | $ 2688 | Muhit M [13] |
| Bhutan | 2018 | 5,050 | | 1% (WHO) | Cataract (53.8%) | $ 3266 | Lepcha NT [11] |
| Narayani Zone, Nepal | 2018 | 4,771 | | 1.2% (WHO) | Cataract (?) | | Pradhan S [17] |
| | | | | 2.7% (SVI) | | | |
| Tibet, China | 2018 | 4,763 | | 1.6% (WHO) | Cataract (39.5%) | | Jiachu D [14] |
| Lahore, Pakistan | 2020 | 2,958 | | 1.2% (WHO) | Cataract (81%) | $ 1597 | Jolley E [16] |
| Tripura, India | 2020 | 4,500 | | 1.5% (WHO) | Cataract (54.5%) | $ 2389 | Marmamula [15] |
| Nepal | 2021 | 5,234 | 50+ | 2.3% (WHO) | 95% cataract + RE | | Shrestha MK [22] |
| Kabul, Afghanistan | 2021 | 3,751 | | 2.4% (WHO) | Cataract (36.8%) | $ 364 | Sapkota [18] |
| Nepal (present study) | 2021 | 32,503 | 50+ | 1.1% (WHO) | Cataract (65.3%) | $ 1337 | - |
| | | | | 1.7% (SVI) | Cataract (83.9%) | | |

The prevalence of blindness was similar to that found in a 2017 RAAB in Bhutan (1.0% [95% CI 0.5–1.4%) but was significantly lower than was found in India in a nationally representative series of RAABs done in 2015–2019 (2.0% [95% CI 1.9–2.1%]) [11, 12]. Blindness prevalence was also lower than estimates from recent subnational RAABs in neighbouring countries, including Bangladesh, Tibet, Tripura state of India, and Lahore (Pakistan) (Table 6) [13–17]. Within the South Asia region, the eye care system in Nepal, which is a geographically challenging country with a modest per capita gross domestic product, has achieved relatively good control of blindness. Inter-sectorial coordination between the government, non-governmental, and private sectors, help from international agencies, an emphasis on primary eye care, and outreach have contributed to this success [8]. Blindness prevalence was much lower than in similarly mountainous Afghanistan, and relative peace and stability have also likely aided progress in Nepal [18].

The prevalence of blindness and any vision impairment <6/12 were both higher in women. This is consistent with population-based surveys from India, Bangladesh, and elsewhere in the region [12, 13, 15, 16, 19–21]. Interventions such as differential treatment pricing, targeted subsidies, and awareness campaigns could help reduce this gap between women and men.

In the federal government system in Nepal, the responsibility for health administration and planning is devolved to the provincial level. Historically, vision impairment in Nepal is yet to be equally distributed across geographic regions. A 2011 survey of people aged 15 years and older in three districts, each representing one of Nepal's three ecological zones (mountainous greater Himalayas, hilly lesser Himalayas, and Terai plains) found that the prevalence of vision impairment was higher in the plains district than the hills or mountains districts [22]. Similarly, we found that Lumbini and Madesh Provinces in the Terai region had a higher prevalence of blindness and any vision impairment compared to the mainly hilly and sparsely populated Sudur Paschim Province in the west and Bagmati Province in Kathmandu valley, where the national capital and largest city is located with more eye health care facilities than other provinces. Lumbini and Madhesh Provinces have the largest provincial populations in Nepal. They border the Indian states of Uttar Pradesh and Bihar, which have the lowest health care indicators among all 38 Indian states [23]. Both Lumbini and Madhesh Provinces have three major eye hospitals performing high-volume, high-quality cataract surgery; however, more services are needed to meet the need for eye care services of the Nepali population in these provinces.

Cataract was the leading cause of blindness (65%), severe vision impairment (84%) and moderate vision impairment (67%). Cataract surgical complications caused 2% of blindness

and 5% of moderate vision impairment; however, population-based surveys reflect the outcomes of surgery done by multiple providers over many years, therefore it is also important to routinely monitor current outcomes through facility level data to promote effective and safe surgery. A separate publication reporting effective cataract surgical coverage and surgical outcomes from this survey series is in preparation. Non-trachomatous corneal opacity (6% of bilateral blindness) was a significant issue in a populace where ocular trauma and corneal infection are common [24]. Investment in prevention and early referral strategies, as well as eye banking, is needed to address corneal blindness, particularly when the additional magnitude of unilateral corneal opacification (not reported here) is considered.

The RAAB survey methodology is not designed to report the prevalence of specific eye conditions. It may overestimate avoidable anterior segment causes of vision impairment (refractive error and cataract) while under-estimating posterior segment conditions [25]. Nevertheless, glaucoma, age-related macular degeneration and the category of 'other posterior segment causes' all contributed to blindness and–along with corneal disease–will require a strategy for sub-specialty expertise to manage the conditions in each province.

## Limitations

RAAB is a rapid survey methodology that uses simple examination equipment and a protocol that prioritizes identifying avoidable causes of vision impairment. RAAB is not designed to estimate the prevalence of certain eye conditions or collect data on risk factors for vision impairment. Three provinces were surveyed using the mobile application developed for RAAB6, while four were surveyed using RAAB7; the RAAB7 app guided data collectors on more or less appropriate responses. However, the questionnaire itself was identical for all provinces. The series of seven surveys took longer to complete than anticipated because data collection was interrupted by the COVID-19 pandemic in 2020. However, we did not find any obvious effect of the pandemic on blindness or VI prevalence in the final province surveyed compared to pre-pandemic results (prevalence was joint lowest among seven here) and believe the provincial findings to be comparable over the duration of the series.

## Conclusion

Cataract was the chief cause of blindness, severe and moderate visual impairment while refractive error was the main cause of mild vision impairment. The prevalence of blindness was more for women than men and varied by province. The Terai, plains region, having the populous Lumbini and Madesh provinces, had a higher prevalence of blindness compared to other five provinces.

## Supporting information

**S1 File. Inclusivity in global research.**
(DOCX)

**S1 Appendix.**
(DOCX)

## Acknowledgments

All the ophthalmic and administrative human resource from Mechi Eye Hospital, Biratnagar Eye Hospital, Sagarmatha Chaudhary Eye Hospital, Gaur Eye Hospital, RM Kedia Eye Hospital, Tilganga Institute of Ophthalmology, Himalaya Eye Hospital, Lumbini Eye Institute, Fateh

Baal Eye Hospital, Geta Eye Hospital and Surkhet Eye Hospital. They provided the human resource (Ophthalmologist, Ophthalmic Assistants and health workers) for the survey. We thank the hospital manager and chief medical director of each hospital for their cooperation and meticulous coordination during the survey. Thanks also to Nabin Rai, Manish Khatiwada and Abhishek Kumar Jha.

Nepal Health Research Council (NHRC) had given ethics committee approval for the study.

IAPB played a technical role in training, executing and consolidating these studies.

## Author Contributions

**Conceptualization:** Sailesh Kumar Mishra, Ranjan Shah, Yuddha Dhoj Sapkota, Matthew Burton.

**Data curation:** Ranjan Shah, Reeta Gurung, Ian McCormick, Brish Bahadur Shahi.

**Formal analysis:** Parikshit Gogate, Yuddha Dhoj Sapkota, Islay Mactaggart, Rajiv Khandekar.

**Funding acquisition:** Sailesh Kumar Mishra.

**Investigation:** Sailesh Kumar Mishra.

**Methodology:** Sailesh Kumar Mishra, Yuddha Dhoj Sapkota, Islay Mactaggart, Ian McCormick, Brish Bahadur Shahi, Matthew Burton.

**Project administration:** Sailesh Kumar Mishra, Ranjan Shah, Yuddha Dhoj Sapkota, Reeta Gurung, Mohan Krishna Shrestha, Islay Mactaggart, Ian McCormick, Brish Bahadur Shahi.

**Resources:** Sailesh Kumar Mishra, Ranjan Shah, Yuddha Dhoj Sapkota.

**Software:** Islay Mactaggart, Ian McCormick.

**Supervision:** Sailesh Kumar Mishra, Yuddha Dhoj Sapkota, Reeta Gurung, Islay Mactaggart, Brish Bahadur Shahi.

**Validation:** Parikshit Gogate, Islay Mactaggart.

**Visualization:** Parikshit Gogate.

**Writing – original draft:** Ranjan Shah, Parikshit Gogate, Yuddha Dhoj Sapkota, Reeta Gurung, Ian McCormick, Rajiv Khandekar.

**Writing – review & editing:** Sailesh Kumar Mishra, Parikshit Gogate, Ian McCormick, Rajiv Khandekar, Matthew Burton.

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
