## [Decision Letter · Decision Letter 0]

20 Oct 2024

PONE-D-24-32405Prevalence of blindness and vision impairment among people 50 years and older in Nepal: a national Rapid Assessment of Avoidable Blindness surveyPLOS ONE

Dear Dr. Gogate,

Thank you for submitting your manuscript to PLOS ONE. After careful consideration, we feel that it has merit but does not fully meet PLOS ONE’s publication criteria as it currently stands. Therefore, we invite you to submit a revised version of the manuscript that addresses the points raised during the review process.

We look forward to receiving your revised manuscript.

Kind regards,

Rohit C. Khanna, MD, MPH

Academic Editor

PLOS ONE

3. Please note that your Data Availability Statement is currently missing a direct link to access each database. If your manuscript is accepted for publication, you will be asked to provide these details on a very short timeline. We therefore suggest that you provide this information now, though we will not hold up the peer review process if you are unable.

4. We note that Figures 1 and 2 in your submission contain [map/satellite] images which may be copyrighted. All PLOS content is published under the Creative Commons Attribution License (CC BY 4.0), which means that the manuscript, images, and Supporting Information files will be freely available online, and any third party is permitted to access, download, copy, distribute, and use these materials in any way, even commercially, with proper attribution. For these reasons, we cannot publish previously copyrighted maps or satellite images created using proprietary data, such as Google software (Google Maps, Street View, and Earth). For more information, see our copyright guidelines: http://journals.plos.org/plosone/s/licenses-and-copyright.

1. You may seek permission from the original copyright holder of Figures 1 and 2 to publish the content specifically under the CC BY 4.0 license. 

5. We note that there is identifying data in the Supporting Information file < RAAB Nepal Start.docx>. Due to the inclusion of these potentially identifying data, we have removed this file from your file inventory. Prior to sharing human research participant data, authors should consult with an ethics committee to ensure data are shared in accordance with participant consent and all applicable local laws.

-Location data

Please remove or anonymize all personal information, ensure that the data shared are in accordance with participant consent, and re-upload a fully anonymized data set. Please note that spreadsheet columns with personal information must be removed and not hidden as all hidden columns will appear in the published file.

Additional Editor Comments:

The manuscript is an important addition to current understanding on prevalence and causes of blindness and VI in Nepal. However, the authors need to address the below comments as well as the comments from reviewers.

1. The title also need to include causes of blindness and VI

2. Why barriers data was collected for <6/18 and did not include for mild VI

3. All the references need to be rechecked once again.

For example, The prevalence of blindness was similar to that found in a 2017 RAAB in Bhutan (1.0% [95% CI 0.5-1.4%) but was significantly lower than was found in India in a nationally representative series of RAABs done in 2015-2019 (2.0% [95% CI 1.9–2.1%]).[11, 12] – The reference is not correct

The last forty years have seen enormous multi-sectorial efforts to improve eye health in Nepal. The number of ophthalmologists has increased from seven in 1981 to 364 in 2019, while the number of optometrists has increased from zero to 857 and ophthalmic assistants increased from zero to 1,246 in the same period.[8] - The reference is not correct

4. Multiple typos which need to be corrected

5. Sentences like - Nepal was one of the first countries to endorse the World Health Organization International Agency for the Prevention of Blindness initiative “VISION 2020 The Right to Sight” and, in 2001, developed its first national eye health plan.[3] – Not sure if this is correct.

Reviewers' comments:

Reviewer's Responses to Questions

**Comments to the Author**

1. Is the manuscript technically sound, and do the data support the conclusions?

Reviewer #1: Yes

Reviewer #2: Yes

Reviewer #3: Yes

2. Has the statistical analysis been performed appropriately and rigorously? 

Reviewer #1: Yes

Reviewer #2: Yes

Reviewer #3: Yes

3. Have the authors made all data underlying the findings in their manuscript fully available?

Reviewer #1: Yes

Reviewer #2: Yes

Reviewer #3: No

4. Is the manuscript presented in an intelligible fashion and written in standard English?

Reviewer #1: Yes

Reviewer #2: Yes

Reviewer #3: Yes

5. Review Comments to the Author

Reviewer #1: The studies have been conducted according to standard protocol, and the manuscript is very clearly written.

Two comments

The survey in the province with least prevalence of blindness and maximum prevalence of blindness have been conducted almost 3 years apart. Do you think that could have influences the results? Looking at the local conditions an opinion could be offered in the discussion section

There is a statement that - 'Both Lumbini and Madhesh Provinces have

three major eye hospitals performing high-volume, high-quality cataract surgery; however,

more services are needed to meet the need for eye care services of the Nepali population in

these provinces as nearly half the hospital workload is catering to patients from neighbouring

Indian states.' - Are there any published data/ reports substantiating this? How could you be sure that it is due to capacity limitation and not issues with access in that province

Reviewer #2: RAAB Nepal – Comments

Thank you for the opportunity to review the manuscript.

Please use consistent font and formatting across the manuscript.

What was the sample size for each province?

How many teams were used in each province?

The duration of data collection varied across the clusters. What are the reasons for this difference?

The duration provided in Table 1 is different from that provided in the text (in red font). Please clarify.

It is unusual to have population-based studies with a response rate close to 100% in a few provinces and also 98% in the overall study. What are the possible reasons for this response rate? Were participants who were not available replaced?

“Eye with VA worse than 6/12 and no obvious anterior segment pathology were dilated”. How many people were dilated for fundus examination? How was the fundus examination done?

How does the results compare with the recent national survey in the India?

The conclusion section is a repetition, please rephrase it.

Reviewer #3: PLOS One comment

General comments

I found this manuscript to be well-written overall. I appreciate all the authors who conducted RAAB in Nepal. Some suggestions as follows will help improve further

On page 4, the last line, …the all-age population prevalence was 0.84% in 1981 and 0.35% in 2011. Further in the round bracket, it is written as extrapolated from estimates among those aged 50 and older.

The authors need to check this. Did the author mean the extrapolation was done for the all-age population from the estimates among people aged 50 and above? In this, how was the extrapolation of the population aged less than 50 years done? The author may look up this.

How many provinces are in Nepal? Was the country divided into zones or provinces? It needs some clarification.

Please reframe or improve the writing of the study objectives.

Explain the survey team structure. Mention any pilot study if conducted.

The timeline does not match between the text on page 7 and Table 1. Check it.

On page 8, the data management, the author may mention the specifications of Android devices if recommended by mRAAB developers. Write the full form of CSV at the first appearance.

The author can share how the data confidentiality was ensured if the Amazon Web Services-based- server was used.

In the results, the author used on many fronts …. ‘significantly higher’…. but I don’t see any statistical value or presentation showing p-value in the text. This may be checked. The author can use an alternative word if it is not associated with a statistically indicated significant value.

On page 20, Check the table 3, when the extrapolated magnitude of female and male are added up, it is not match with total extrapolated figure. The author can check it. Other tables seems okay, but re-check it.

The author may present an infographic on the trend of the prevalence of blindness and visual impairment since inception of the national survey in Nepal. This will make readers interested to understand the changes that happened over the years.

Please check the resolution of Figure. Please re-look at the guidelines if any.

6. PLOS authors have the option to publish the peer review history of their article (what does this mean?). If published, this will include your full peer review and any attached files.

Reviewer #1: **Yes: **Dr Shalinder Sabherwal

Reviewer #2: No

Reviewer #3: **Yes: **Suraj Singh Senjam

---

## [Author Response · Author response to Decision Letter 0]

22 Nov 2024

2. Please include a complete copy of PLOS’ questionnaire on inclusivity in global research in your revised manuscript. Our policy for research in this area aims to improve transparency in the reporting of research performed outside of researchers’ own country or community. The policy applies to researchers who have travelled to a different country to conduct research, research with Indigenous populations or their lands, and research on cultural artefacts. The questionnaire can also be requested at the journal’s discretion for any other submissions, even if these conditions are not met. Please find more information on the policy and a link to download a blank copy of the questionnaire here: https://journals.plos.org/plosone/s/best-practices-in-research-reporting. Please upload a completed version of your questionnaire as Supporting Information when you resubmit your manuscript. The PLOS questionnaire on inclusivity in global research has been added to the submission.

3. Please note that your Data Availability Statement is currently missing a direct link to access each database. If your manuscript is accepted for publication, you will be asked to provide these details on a very short timeline. We therefore suggest that you provide this information now, though we will not hold up the peer review process if you are unable.

The datasets are owned by Nepal Netra Jyoti Sangh and are available to download for registered users of the Rapid Assessment of Avoidable Blindness (RAAB) repository via https://www.raab.world/country-profiles/nepal (registration is free). Sudurpashchim Province https://www.doi.org/10.17037/RAABDATASET.00000385; Gandaki Province https://www.doi.org/10.17037/RAABDATASET.00000377; Madhesh Province https://www.doi.org/10.17037/RAABDATASET.00000379; Koshi Province https://www.doi.org/10.17037/RAABDATASET.00000378; Bagmati Province https://www.doi.org/10.17037/RAABDATASET.00000360; Karnali Province https://www.doi.org/10.17037/RAABDATASET.00000361; Lumbini Province https://www.doi.org/10.17037/RAABDATASET.00000349. The license restricts use of the data for commercial purposes. This restriction has been imposed in accordance with the research ethics approved institutionally by LSHTM and by partner institutions conducting RAABs.

4. We note that Figures 1 and 2 in your submission contain [map/satellite] images which may be copyrighted. All PLOS content is published under the Creative Commons Attribution License (CC BY 4.0), which means that the manuscript, images, and Supporting Information files will be freely available online, and any third party is permitted to access, download, copy, distribute, and use these materials in any way, even commercially, with proper attribution. For these reasons, we cannot publish previously copyrighted maps or satellite images created using proprietary data, such as Google software (Google Maps, Street View, and Earth). For more information, see our copyright guidelines: http://journals.plos.org/plosone/s/licenses-and-copyright.

The maps in Figures 1 and 2 were created in R by one of the co-authors (IM) using open data from the UN Office for the Coordination of Humanitarian Affairs (OCHA) hosted on Humanitarian Data Exchange (https://data.humdata.org/dataset/cod-ab-npl). The data are licenced under a CC BY 3.0 IGO licence (https://creativecommons.org/licenses/by/3.0/igo/), therefore, we do not believe that we require written permission to produce the maps for these figures. Please advise.

1. You may seek permission from the original copyright holder of Figures 1 and 2 to publish the content specifically under the CC BY 4.0 license. 

5. We note that there is identifying data in the Supporting Information file < RAAB Nepal Start.docx>. Due to the inclusion of these potentially identifying data, we have removed this file from your file inventory. Prior to sharing human research participant data, authors should consult with an ethics committee to ensure data are shared in accordance with participant consent and all applicable local laws.

Individual patient data has not been shared or shown at any point in this manuscript. 

-Location data

Please remove or anonymize all personal information, ensure that the data shared are in accordance with participant consent, and re-upload a fully anonymized data set. Please note that spreadsheet columns with personal information must be removed and not hidden as all hidden columns will appear in the published file.

Additional Editor Comments:

The manuscript is an important addition to current understanding on prevalence and causes of blindness and VI in Nepal. However, the authors need to address the below comments as well as the comments from reviewers.

Thank you for your positive feedback and your time in reviewing this manuscript.

1. The title also need to include causes of blindness and VI. 

Thank you. This has been amended as per reviewer’s suggestions:

“Prevalence and causes of blindness and vision impairment among people 50 years and older in Nepal: a national Rapid Assessment of Avoidable Blindness survey”

2. Why barriers data was collected for <6/18 and did not include for mild VI. 

This is how it was planned in 2018-19. This manuscript does not present the barriers data. 

3. All the references need to be rechecked once again. For example, The prevalence of blindness was similar to that found in a 2017 RAAB in Bhutan (1.0% [95% CI 0.5-1.4%) but was significantly lower than was found in India in a nationally representative series of RAABs done in 2015-2019 (2.0% [95% CI 1.9–2.1%]).[11, 12] – The reference is not correct

The last forty years have seen enormous multi-sectorial efforts to improve eye health in Nepal. The number of ophthalmologists has increased from seven in 1981 to 364 in 2019, while the number of optometrists has increased from zero to 857 and ophthalmic assistants increased from zero to 1,246 in the same period.[8] - The reference is not correct

Thank you for highlighting these, we have corrected the references.

4. Multiple typos which need to be corrected. 

A complete spell check has been done again.

5. Sentences like - Nepal was one of the first countries to endorse the World Health Organization International Agency for the Prevention of Blindness initiative “VISION 2020 The Right to Sight” and, in 2001, developed its first national eye health plan.[3] – Not sure if this is correct. 

This has been edited and rephrased. Nepal endorsed the WHO-IAPB initiative ‘Vision 2020: The Right to Sight’ and in 2001 developed its first national eye health plan. 

Reviewer #1: The studies have been conducted according to standard protocol, and the manuscript is very clearly written.

Thank you for your positive feedback and your time to review this manuscript.

Two comments

The survey in the province with least prevalence of blindness and maximum prevalence of blindness have been conducted almost 3 years apart. Do you think that could have influences the results? Looking at the local conditions an opinion could be offered in the discussion section. 

Thank you for raising this interesting point. Covid pandemic happened during the study period and that stretched the timelines as community surveys and door to door campaigning was prohibited for a time. The prevalence in Lumbini (completed 2018) was 1.8% (1.3-2.2) while the prevalence in Sudurpaschim (completed 2021) was 0.7% (0.4-0.9). The prevalence in Sudurpaschim in 2021 was the same as Bagmati (completed 2019) and similar to Koshi (completed 2019) so is not an outlier among the seven provincial results. Given the disruption that Covid caused to service provision in 2020-21, we might anticipate any temporal trend to affect a difference in the opposite direction to what was found here, all other things being equal. Therefore, we think the estimates from the range of years 2018-2021 are reasonably comparable.

We have added the following underlined text to the limitations section:

“The series of seven surveys took longer to complete than anticipated because data collection was interrupted by the COVID-19 pandemic in 2020. However, we did not find any obvious effect of the pandemic on blindness or VI prevalence in the final province surveyed compared to pre-pandemic results (prevalence was joint lowest among seven here) and believe the provincial findings to be comparable over the duration of the series.”

There is a statement that - 'Both Lumbini and Madhesh Provinces have

three major eye hospitals performing high-volume, high-quality cataract surgery; however, more services are needed to meet the need for eye care services of the Nepali population in these provinces as nearly half the hospital workload is catering to patients from neighbouring Indian states.' - Are there any published data/ reports substantiating this? How could you be sure that it is due to capacity limitation and not issues with access in that province. 

The sentence has been re-worded as: 'Both Lumbini and Madhesh Provinces have

three major eye hospitals performing high-volume, high-quality cataract surgery; however, more services are needed to meet the need for eye care services of the Nepali population in these provinces. 

Reviewer #2: RAAB Nepal – Comments

Thank you for the opportunity to review the manuscript. Please use consistent font and formatting across the manuscript.

Thank you for your time and your input.

What was the sample size for each province?

This information is provided in tables 1 (target sample size per province) and 2 (numbers enrolled and examined per province).

How many teams were used in each province?

Each province had three teams, with each team comprising an Ophthalmologist, an Ophthalmic Assistant, and a motivator/data enumerator/cluster informer (Eye Health Worker). Additionally, a hospital-based logistics coordinator was assigned to support the teams. Two teams were deployed at all times, while one backup team was available as needed.

The duration of data collection varied across the clusters. What are the reasons for this difference? 

Covid pandemic happened during the study period and that stretched the timelines as community surveys and door to door campaigning was prohibited during the pandemic. Also there were differences in sample size, at times there was a limitation in the number of personnel and the travel/logistics challenges were different for the hilly and mountainous regions. 

The duration provided in Table 1 is different from that provided in the text (in red font). Please clarify. The data in text, red font, refers to the duration of approval given by the ethics committee while the table shows the actual duration of data collection. 

NHRC- Duration of Approval

Province 01 (Koshi) from 5 June 2019 to 30th September 2019; in Province 02 (Madhesh) from 18 December 2019 to 18 December 2020; Province 03 (Bagmati) from 12 June 2019 to December 2019; in Province 04 (Gandaki) from 05 June 2019 to October 2019; in Province 05 (Lumbini) from 03 October 2018 to December 2018; in Province 06 (Karnali) from 03 January 2019 to 02 January 2020 and in Province 07 (Sudur Paschim) from19 July 2020 to 19 July 2021. We have presented only the data collection dates in the re-submission to avoid confusion.

Actual Data Collection happened within the duration of Approval

Table 1. Sampling and data collection information for each provincial RAAB survey in Nepal 2018-2021.

Province Sampling information Data collection

 Estimated prevalence of blindness Population 50 years and older (2011 census) Sample size required Number of clusters (size 35) Data collection period RAAB version used

Koshi 3.3% 743,329 4360 125 June - September 2019 6

Madhesh 3.5% 760,911 4109 118 Jan- April 2020 7

Bagmati 

---

## [Decision Letter · Decision Letter 1]

27 Dec 2024

PONE-D-24-32405R1Prevalence and causes of blindness and vision impairment among people 50 years and older in Nepal: a national Rapid Assessment of Avoidable Blindness surveyPLOS ONE

Dear Dr. Gogate,

Thank you for submitting your manuscript to PLOS ONE. After careful consideration, we feel that it has merit but does not fully meet PLOS ONE’s publication criteria as it currently stands. Therefore, we invite you to submit a revised version of the manuscript that addresses the points raised during the review process.

We look forward to receiving your revised manuscript.

Kind regards,

Rohit C. Khanna, MD, MPH

Academic Editor

PLOS ONE

Journal Requirements:

Additional Editor Comments:

Thanks for the revision, however there are still some comments to be addressed.

1. In the introduction, the authors shared that all-age population prevalence of blindness was extrapolated from the prevalence data of 50 years and older. I think this is not possible at all without employing or knowing the data of people less than 50 years and younger. Hence, would suggest to remove this extrapolation.

2. In abstract statement like 'The prevalence of blindness and any vision impairment were both higher in women than men (1.3% [95% CI 1.1-1.5%] vs 0.9% [95% CI 0.7-1.0%]).' However the data is shown only for blindness. Also statement like 'Cataract was the leading cause of blindness, severe vision impairment and moderate vision impairment while refractive error was the leading cause of mild vision impairment.' is not supported by data in abstract

3. References are still not appropriate. Would appreciate if the authors can go through each reference and ensure they are the right reference and in right place.

Reviewers' comments:

Reviewer's Responses to Questions

**Comments to the Author**

1. If the authors have adequately addressed your comments raised in a previous round of review and you feel that this manuscript is now acceptable for publication, you may indicate that here to bypass the “Comments to the Author” section, enter your conflict of interest statement in the “Confidential to Editor” section, and submit your "Accept" recommendation.

Reviewer #1: All comments have been addressed

Reviewer #2: All comments have been addressed

Reviewer #3: All comments have been addressed

2. Is the manuscript technically sound, and do the data support the conclusions?

Reviewer #1: Yes

Reviewer #2: Yes

Reviewer #3: Yes

3. Has the statistical analysis been performed appropriately and rigorously? 

Reviewer #1: Yes

Reviewer #2: Yes

Reviewer #3: N/A

4. Have the authors made all data underlying the findings in their manuscript fully available?

Reviewer #1: Yes

Reviewer #2: (No Response)

Reviewer #3: No

5. Is the manuscript presented in an intelligible fashion and written in standard English?

Reviewer #1: Yes

Reviewer #2: Yes

Reviewer #3: Yes

6. Review Comments to the Author

Reviewer #1: (No Response)

Reviewer #2: All comments addressed. However, I suggest to include the number of people who had fundus examination done.

Reviewer #3: In the results,

So, the words "significant higher" is not meant to be statistically significant since no hypothesis test was done during the analysis. Maybe the author should consider adding one or tow lines on this in the analysis section so no confusion thereby.

7. PLOS authors have the option to publish the peer review history of their article (what does this mean?). If published, this will include your full peer review and any attached files.

Reviewer #1: **Yes: **Dr Shalinder Sabherwal

Reviewer #2: No

Reviewer #3: **Yes: **suraj singh senjam

---

## [Author Response · Author response to Decision Letter 1]

30 Dec 2024

1. In the introduction, the authors shared that all-age population prevalence of blindness was extrapolated from the prevalence data of 50 years and older. I think this is not possible at all without employing or knowing the data of people less than 50 years and younger. Hence, would suggest to remove this extrapolation. This extrapolation has been published earlier: Sapkota YD, Limburg H. The epidemiology of blindness in Nepal: 2012. Kathmandu: Nepal Netra Jyoti Sangh. 2012 May 30 https://www.iapb.org/wp-content/uploads/Epidemiology-of-Blindness-Nepal.pdf

2. In abstract statement like 'The prevalence of blindness and any vision impairment were both higher in women than men (1.3% [95% CI 1.1-1.5%] vs 0.9% [95% CI 0.7-1.0%]).' However the data is shown only for blindness. Also statement like 'Cataract was the leading cause of blindness, severe vision impairment and moderate vision impairment while refractive error was the leading cause of mild vision impairment.' is not supported by data in abstract. This has been changed as per reviewer advice. The following sentence has been added to the results section of the abstract: . Cataract was also the leading cause of severe vision impairment (83.9%) and moderate vision impairment (66.8%), while refractive error was the leading cause of mild vision impairment (66.5%)

3. References are still not appropriate. Would appreciate if the authors can go through each reference and ensure they are the right reference and in right place. The references have been edited. Reference 12 : Neena J, Rachel J, Praveen V, Murthy GV, RAAB India Study Group. Rapid assessment of avoidable blindness in India. PloS One 2008;3:e2867. Has been replaced by Vashist P, Senjam SS, Gupta V, Gupta N, Shamanna BR, Wadhwani M, et al. Blindness and visual impairment and their causes in India: Results of a nationally representative survey. PLoS ONE 2022; 17(7): e0271736.

Reviewer #2: All comments addressed. However, I suggest to include the number of people who had fundus examination done. In the examined sample, all those who did not have a dense mature cataract had their fundus examined, there were 17,935 persons of 50 and more years of age who underwent fundus examination.

Reviewer #3: In the results,

So, the words "significant higher" is not meant to be statistically significant since no hypothesis test was done during the analysis. Maybe the author should consider adding one or tow lines on this in the analysis section so no confusion thereby. We have stated in the results section: The prevalence of blindness was significantly higher among females (1.3% [95% CI 1.1 - 1.5%]) than males (0.9% [95% CI 0.7 - 1.0%]); the difference in vision impairment prevalence by sex was not significantly higher at any other threshold of vision impairment (table 3). This has been documented in table 3. Would the reviewer/editorial team advise how this needs to be modified?

---

## [Editor Report · Decision Letter 2]

2 Jan 2025

Prevalence and causes of blindness and vision impairment among people 50 years and older in Nepal: a national Rapid Assessment of Avoidable Blindness survey

PONE-D-24-32405R2

Dear Dr. Gogate

We’re pleased to inform you that your manuscript has been judged scientifically suitable for publication and will be formally accepted for publication once it meets all outstanding technical requirements.

Kind regards,

Rohit C. Khanna, MD, MPH

Academic Editor

PLOS ONE

---

## [Editor Report · Acceptance letter]

12 Jan 2025

PONE-D-24-32405R2 

PLOS ONE

Dear Dr. Gogate, 

I'm pleased to inform you that your manuscript has been deemed suitable for publication in PLOS ONE. Congratulations! Your manuscript is now being handed over to our production team.

Kind regards, 

on behalf of

Dr. Rohit C. Khanna 

Academic Editor

PLOS ONE